# Insights into the Role of the Microbiota and of Short-Chain Fatty Acids in Rubinstein–Taybi Syndrome

**DOI:** 10.3390/ijms22073621

**Published:** 2021-03-31

**Authors:** Elisabetta Di Fede, Emerenziana Ottaviano, Paolo Grazioli, Camilla Ceccarani, Antonio Galeone, Chiara Parodi, Elisa Adele Colombo, Giulia Bassanini, Grazia Fazio, Marco Severgnini, Donatella Milani, Elvira Verduci, Thomas Vaccari, Valentina Massa, Elisa Borghi, Cristina Gervasini

**Affiliations:** 1Department of Health Sciences, Università degli Studi di Milano, 20142 Milan, Italy; elisabetta.difede@unimi.it (E.D.F.); emerenziana.ottaviano@unimi.it (E.O.); paolo.grazioli@unimi.it (P.G.); camilla.ceccarani@unimi.it (C.C.); chiara.parodi@unimi.it (C.P.); elisaadele.colombo@unimi.it (E.A.C.); giulia.bassanini@unimi.it (G.B.); elvira.verduci@unimi.it (E.V.); valentina.massa@unimi.it (V.M.); elisa.borghi@unimi.it (E.B.); 2Institute of Biomedical Technologies, Italian National Research Council, Segrate, 20054 Milan, Italy; marco.severgnini@itb.cnr.it; 3Department of Biosciences, Università degli Studi di Milano, 20133 Milano, Italy; antonio.galeone@unimi.it (A.G.); thomas.vaccari@unimi.it (T.V.); 4Tettamanti Research Center, Department of Pediatrics, Università degli Studi di Milano-Bicocca, MBBM Foundation/San Gerardo Hospital, 20900 Monza, Italy; grazia.fazio@unimib.it; 5Fondazione IRCCS Ca’ Granda Ospedale Maggiore Policlinico, 20122 Milan, Italy; donatella.milani@policlinico.mi.it; 6Department of Pediatrics, Vittore Buzzi Children’s Hospital, University of Milan, 20154 Milan, Italy; 7“Aldo Ravelli” Center for Neurotechnology and Experimental Brain Therapeutics, Università degli Studi di Milano, 20142 Milan, Italy

**Keywords:** Rubinstein–Taybi syndrome, butyrate, microbiota, HDACi, histones

## Abstract

The short-chain fatty acid butyrate, produced by the gut microbiota, acts as a potent histone deacetylase (HDAC) inhibitor. We assessed possible ameliorative effects of butyrate, relative to other HDAC inhibitors, in in vitro and in vivo models of Rubinstein–Taybi syndrome (RSTS), a severe neurodevelopmental disorder caused by variants in the genes encoding the histone acetyltransferases CBP and p300. In RSTS cell lines, butyrate led to the patient-specific rescue of acetylation defects at subtoxic concentrations. Remarkably, we observed that the commensal gut microbiota composition in a cohort of RSTS patients is significantly depleted in butyrate-producing bacteria compared to healthy siblings. We demonstrate that the effects of butyrate and the differences in microbiota composition are conserved in a *Drosophila melanogaster* mutant for CBP, enabling future dissection of the gut–host interactions in an in vivo RSTS model. This study sheds light on microbiota composition in a chromatinopathy, paving the way for novel therapeutic interventions.

## 1. Introduction

Gene expression regulation is mediated by tightly balanced epigenetic mechanisms involving histone modifications, such as acetylation and methylation. Correct histone acetylation levels on lysine residues are fundamental for several physiological processes, including embryonic development [1,2]. Two classes of functionally antagonistic enzymes, the acetyltransferases (HAT) and deacetylases (HDAC), are known to modulate histone acetylation levels [3]. Histones hypoacetylation has been associated with alterations in synaptic plasticity, neuronal survival/regeneration, memory formation [4], while defects in epigenetic components acting on acetylation status cause several neurodevelopmental/malformation syndromes [5]. Among them, Rubinstein–Taybi syndrome (RSTS, OMIM #180849, #613684) is a rare (1:125,000) autosomal-dominant disease characterized by a wide and heterogeneous spectrum of clinical signs [6]. These include intellectual disability of variable entity (ranging from mild to severe), postnatal growth deficiency, distinctive dysmorphisms, skeletal abnormalities (such as typical broad thumbs and large toes), multiple congenital anomalies (e.g., heart defects), and several additional clinical problems such as constipation [7]. Albeit the growth in height is constantly reduced in RSTS patients, growth in weight is reduced neonatally and in the first infancy, but at puberty an excessive weight gain is observed [8].

Most RSTS cases are caused by de novo monoallelic variants of one of two highly conserved genes: *CREBBP*, located at 16p13.3, coding for the CREB (cAMP response element-binding protein) binding protein (CBP) and *EP300*, mapping at 22q13.2, coding for the E1A-associated protein p300. *CREBBP* is found mutated in >50% RSTS patients, while *EP300* gene mutations have been described in a minor fraction of patients [9].

Somatic mutations in *CREBBP* and *EP300* are reported in different benign and malignant tumors, and an association between RSTS patients and tumor development has been investigated. This disorder is related to an increased risk of malignancies up to 5%, in particular involving cutaneous, hematological, and brain tumors such as pilomatrixoma, leukemia, and meningioma, respectively [10,11].

CBP and p300 have ubiquitously expressed paralog proteins belonging to the lysine acetyl transferases (HAT) family [12]. CBP and p300 act as co-factors for transcription and are required in multiple pathways controlling cell growth, DNA repair, cell differentiation, and tumor suppression [13,14,15,16]. Their acetylation of target histone tails enables the opening of chromatin, thus promoting gene expression [13,15,17].

In recent years, a novel class of compounds, termed HDAC inhibitors (HDACi), has been used to increase histone acetylation in different pathologies [18,19]. Preliminary studies testing the efficiency of HDACi to revert acetylation defects in RSTS lymphoblastoid cell lines (LCLs) supported the hypothesis that RSTS is caused by an acetylation imbalance [20]. Animal model studies introduced the idea that the chromatin alterations observed in RSTS could be reverted [21].

It has been demonstrated that protein acetylation can be modulated by the commensal microbial community (microbiota from here on) [22]. In fact, short-chain fatty acids (SCFAs), such as acetate, propionate, and butyrate, the most abundant products of anaerobic fermentation of the gut microbiota, can act as HDACi. Among SCFAs, butyrate is exclusively produced by commensal microorganisms and widely reported for its epigenetic activity, making it the most potent HDACi among natural compounds [23,24]. However, the role of butyrate or the composition of the microbiota in RSTS have not been investigated. Altered gut microbiota could itself affect the endogenous levels of SCFAs in patients, it could participate in their typical RSTS growth trend, characterized by a deficit in infancy and excessive weight gain after puberty, and/or it could contribute to the comorbidities often associated with RSTS, such as gastrointestinal discomfort [8].

On these premises, in the present study, we compared butyrate to other HDACi molecules in vitro on lymphoblastoid cell lines (LCLs) derived from RSTS patients. We have found it effective in modulating the acetylation impairment associated with reported CBP/p300 defects [20]. Remarkably, we also find that the microbiota of RSTS patients is poor in SCFA-producing bacteria, perhaps further contributing to acetylation imbalance. Finally, using *Drosophila melanogaster*, we model the effects of butyrate and microbiota alterations for future in vivo studies. Our work points to the importance of the microbiome in the pathogenesis and treatment of ultra-rare diseases.

## 2. Results

### 2.1. HDACi Exposure Counteracts Acetylation Imbalance in RSTS Lymphoblastoid Cell Lines (LCLs)

To investigate the effect of SCFAs as HDACi in RSTS, we exposed LCLs derived from eight patients, four with *CREBBP* and four with *EP300* mutations (Appendix A) and seven healthy donors (HD) to sodium butyrate (NaB), and we compared the effect to that of three other HDACi: trichostatin A (TSA), suberoylanilide hydroxamic acid (SAHA), and valproic acid (VPA) (Appendix A). By AlphaLISA^®^ assay, we analyzed the acetylation levels of lysine 27 of histone H3 (H3K27ac) in LCLs upon three different conditions: HDACi treatments, exposure to the vehicle (DMSO or H_2_O), and untreated cells (Figure 1).

All the compounds succeeded in boosting histone acetylation in RSTS LCLs compared to healthy donor (HD) LCLs, with VPA exposure resulting highly significant (*p* < 0.01). This increment was particularly manifest in patient derived LCLs compared to untreated samples (Figure 1a).

We also observed that HDACi compounds induced a variable acetylation response, in a patient-specific manner when compared to treated HD LCLs (Figure 1b). As shown in Figure 1b, treatment with TSA 2 µM boosted acetylation levels significantly in LCLs RSTS 176 (*p* < 0.001), RSTS 25 (*p* < 0.001) and RSTS 39 (*p* < 0.01), while SAHA 2 µM showed highly significant effect on RSTS 176 and RSTS 25 (*p* < 0.001). VPA 2mM treatment particularly increased H3K27ac of RSTS 114, RSTS 120, RSTS 176, and RSTS 54 (*p* < 0.001), while exposure to NaB 5 mM significantly affected acetylation of RSTS 176 and RSTS 39 (*p* < 0.001), RSTS 54, and RT010-15 (*p* < 0.05).

Of note, when analyzing specific RSTS patient-derived LCLs response to HDACi compared to the relative untreated conditions, we observed that at least one HDACi significantly boosted acetylation and that RSTS-LCLs response varied among different drug treatments (Figure 1c and Appendix A). These data indicate that SCFAs as NaB show patient-specific acetylation increases, as is the case of other HDACi.

Considering HDACi applications as anticancer drugs for their role in cell cycle arrest, cell death, and immune-mediated mechanisms [25], we studied NaB and other HDACi effects on cell proliferation and apoptosis, performing Ki67 and Tunel assays upon exposure of LCLs with HDACi (Appendix A). For both assays, we did not observe a significant correlation with H3K27 acetylation (Appendix A), indicating that the HDACi rescue did not impair cell cycle progression or promote cell death under the experimental conditions used.

### 2.2. RSTS Patients Are Depleted in the Major Butyrate-Producer Faecalibacterium spp.

Because in vitro evaluation indicated that SCFAs can act as HDACi in RSTS, we focused on investigating the production of SCFAs by the commensal microbiota of the patients. To this end, we enrolled 23 RSTS subjects (mean age 10.2 ± 6.4 years; 12 females) and 16 healthy siblings (healthy donors, HD), mean age 12.7 ± 7.2 years; 6 females), as a control group to minimize environmental factors having a well-recognized role on gut microbiota. The dietary survey revealed no differences in intake of macronutrients. However, energy intake was lower in RSTS subjects when compared to HD controls (*p* = 0.0054). Nutritional parameters are detailed in the relative Appendix A [26].

Microbiota profiling was performed through 16S rRNA gene-targeted sequencing. After quality filtering processes, we obtained a mean count of 90,759 reads per sample. The alpha-diversity analysis of the gut microbiota showed no significant differences between RSTS and HD fecal samples in terms of richness (Observed species: *p* = 0.255; Phylogenetic Diversity (PD) whole tree: *p* = 0.279—see Figure 2a) and of richness and evenness (Chao1: *p* = 0.151; Shannon: *p* = 0.287—see Appendix A). Beta-diversity analysis, instead, showed that RSTS fecal microbiota differed significantly from that of HD according to both unweighted (*p* = 0.013) and weighted (*p* = 0.022) Unifrac distances (beta-diversity, Figure 2b).

The overall composition of the intestinal microbiota (Figure 2c; Appendix A) showed a decreased relative abundance of the Firmicutes phylum (58.5% in RSTS vs. 73.4% in HD, *p* = 0.019), of the *Ruminococcaceae* family (32.2% vs. 41.9%, *p* = 0.049) and of the SCFA-producing *Faecalibacterium* spp. (3.3% vs. 9.8% in HD, *p* = 0.001) in RSTS subjects. On the other hand, RSTS samples showed an enrichment in the *Bacteroidaceae* family and in *Bacteroides* spp. (21.1% vs. 10.3%, *p* = 0.021), as well as in *Oscillospira* spp. (5.1% vs. 2.4% in HD, *p* = 0.007). Matched-pair analysis (Wilcoxon signed-rank test), performed on RSTS/sibling pairs showed a significant and environment-independent decrease (*p* = 0.0021) in *Faecalibacterium* spp. (Figure 2d).

Afterward, we directly measured SCFA abundance in patient fecal samples. A decreasing trend in butyrate content was observed (4.13 ± 1.40 vs. 5.14 ± 1.79 mg/g feces, *p* = 0.0741, Mann–Whitney test), whereas acetate, propionate, and branched-chain fatty acids (iso-butyrate and iso-valerate) concentrations were similar (*p* = 0.194, *p* = 0.874, *p* = 0.786, and *p* = 0.467, respectively). These data indicate that RSTS patient microbiota produces low levels of endogenous butyrate.

### 2.3. HDACi Exposure Leads to Partial Rescue of RSTS Phenotype in Drosophila CBP Mutants

To establish an in vivo model to study the effect of SCFAs and the microbiome on RSTS pathogenesis, we analyzed *Drosophila melanogaster* mutants in the CBP homolog *nejire (nej)*. Homozygous *nej* loss of function results in abnormal embryonic development leading to early lethality [27,28]. In particular, mutations in *nej* lead to defects in morphology caused by misregulation of wingless (wg) and other signaling pathways at stage 9 of embryonic development [27]. To test whether NaB, or VPA as a control HDACi, upon supplementation in the food could modulate emergence of *nej* mutant defects in vivo, we collected embryos deposed by females fed with the drugs and assessed their morphology at stages 8–12 (Figure 3 and Appendix A).

Compared to *nej^+/−^* siblings, we have found that the majority of *nej^−/−^* embryos die between stages 8 and 10 with the reported twisting of embryo morphology [28] (Figure 3a, quantification in Figure 3b). Upon NaB or VPA treatment, *nej^−/−^* embryos display a partially rescued embryonic development, statistically significant at stage 10 (*p* < 0.01). Importantly, both treatments extended the survival of *nej^−/−^* embryos to stage 12, although twisting of the embryos is morphologically visible as bottleneck and cracks (Figure 3 and Appendix A). The reported twisting phenotype in treated *nej^−/−^* embryos is often less dramatic than in untreated *nej^−/−^* embryos allowing segmentation. Notably, both VPA and NaB treatment did not show any developmental delay or morphological defects in control embryos (data not shown). Altogether, these results indicate that NaB acts as a HDACi in vivo to ameliorate the developmental defects associated with acute loss of CBP homologs.

### 2.4. The Fly Gut Microbiota of Heterozygous Drosophila CBP Mutants

Because RSTS patients possess a defective microbiota, we analyzed that of heterozygous *nej* (*nej^+/−^*) flies. In contrast to *nej^−/−^* embryos, *nej^+/−^* animals progress to adulthood and display no overt defects. However, they have been shown to reveal genetic interaction with genes involved in developmental processes regulated by CBP [28]. Hence, we reasoned that adult *nej^+/−^* animals could be used to investigate the fly gut microbiota.

A total of 10 samples were obtained from the two experimental groups (*yw* and *nej^+/−^*), with five replicates each (three dissected guts in each replicate, with a total of 15 flies per group).

Alpha diversity metrics (Figure 4a) revealed that *nej* gut microbiota were enriched in low abundant species (Chao1 metric, *p* = 0.005), whereas the phylogenetic diversity between groups showed no significant differences (PD whole tree metric, *p* = 0.668).

Similarly, beta-diversity analysis (Figure 4b) highlighted clear discrimination between *yw* and *nej* microbial communities considering both abundant and rare species within the microbiota (unweighted Unifrac distance, *p* = 0.007),

As for the bacterial composition, the Firmicutes phylum was found more abundant in the *nej* flies (64.7% vs. 53.9% in *yw*), with a concurrent reduction of Proteobacteria (31.6 vs. 41.1% in *yw*). The taxonomic analysis revealed a remarkable increase in *Lactobacillaceae* (58.8% in *nej* vs. 7.5% in *yw*; *p* = 0.0000254), and a profound decrease in the *Enterococcaceae* family (0.2% vs. 12.9% in *yw*; *p* = 0.00784) (Figure 4c). These results were confirmed at genus level, with a significant enrichment in *Lactobacillus* (58.4% in *nej* vs. 3.4% in *yw*; *p* = 0.00000806) and a depletion in *Enterococcus* (0% vs. 3% in *yw*; *p* = 0.00772) (data not shown).

## 3. Discussion

Current therapeutic approaches for RSTS patients are not targeted towards modulation of acetylation and are rather directed towards alleviating clinical symptoms and preventing possible known comorbidities. Common interventions include behavioral support and surgical procedures for the correction of orthopedic or cardiac malformations. In this context, exploring the effects of drugs with an established and specific molecular function in acetylation in preclinical studies is a fundamental step to devise effective future therapy. In the case of RSTS, HDACi are available and have already been used to treat neurological disorders [29]. Hence, the presented work explored the effects of HDAC inhibition in RSTS experimental models, focusing on natural SCFAs such as butyrate.

Treatment with a number of HDACi used in cancer therapy [30,31,32,33] showed a general boost in histone acetylation levels in RSTS patient-derived cell lines. Such increment was significant in a patient-specific manner. Each line derived from different RSTS patients with discrete pathogenic variants responded differently to tested compounds. These data provide evidence for HDACi’s ability to restore acetylation levels in an in vitro model of RSTS, strongly pointing to the possibility of future therapies tailored to individual patients, a central tenet of personalized medicine.

HDACi are tested in oncology trials for their ability to stop tumor cell proliferation by inducing selected and dose-dependent apoptosis [34]. Thus, our observation that selected HDACi dosing does not modulate cell proliferation or death in RSTS cells indicates that HDACi can boost acetylation at a sub-toxic concentration, at least in vitro.

Due to the HDACi function exerted by SCFAs, they have been tested in clinical trials for recurrent malignant gliomas or myelodysplastic syndrome and they have also been studied in vitro against Burkitt lymphoma, primary acute myeloid leukemia, retinoblastoma, medulloblastoma, prostate cancer, and hepatocellular and colon carcinoma [35,36].

Importantly, NaB, a natural SCFA with potent HDACi activity [37], performed on par with other HDACi in RSTS cells. We propose that it should be evaluated for the treatment of patients, considering that it is present in human diets as a product of the human microbiota or a well-tolerated supplement.

To investigate whether butyrate is normally produced by commensal bacteria of RSTS patients, we sequenced the V3–V4 hypervariable regions of the 16S rRNA genes and measured SCFAs as main microbial metabolites. While we scored no differences in nutritional parameters and in microbiota biodiversity in patients versus healthy siblings, we observed a distinct and highly interesting microbial signature characterized by loss of the butyrate-producer genus *Faecalibacterium* [38] in the microbiota of RSTS patients. Such change could participate in the syndrome comorbidity insurgence, perhaps in the gut, and further study should now aim at elucidation of possible genetic-microbial additive negative effects. Interestingly, a reduction of *Faecalibacterium* relative abundance was also reported in patients affected by Rett syndrome, autism spectrum disorder, and down syndrome, suggesting a shared microbiota signature in neurodevelopmental disorders [39].

A recent study reported that a ketogenic diet, highly impacting microbiota composition and metabolism [40], induces the production of deacetylase inhibitors in a mouse model of Kabuki syndrome (OMIM# 147920, # 300867), a rare disease sharing traits and histone modification defect with Rubinstein–Taybi syndrome [41]. Thus, nutritional interventions could also aim at rebalancing the microbiota of RSTS patients. In line with this approach, brain functions and behavior appear more and more influenced, through a bottom-up modulation, by the gut microbiota [42].

It is worth noting that carbohydrates, and in particular fermentable dietary fibers, the most important substrates for short-chain fatty acid production [43], were very similar in RSTS patients and healthy siblings, inconsistent with the reduction in the relative abundance of *Faecalibacterium* genus, or with the lower butyrate fecal concentration. Nutritional recommendations for RSTS comorbidity management are currently lacking as no studies focused on this. Our findings represent a starting point for the evaluation of specific nutritional regimens, which could also shed light on the basis of observed differences.

Studies with *nej* flies underscored the role of CBP during embryogenesis and as coactivators of critical signaling pathways involved in patterning [28,44]. In support of results from our in vitro RSTS model, we observed that rearing flies with food supplemented with NaB lead to partial rescue of embryogenesis and patterning, suggesting that nutritional intervention may ameliorate RSTS traits in vivo.

However, even if the Drosophila model has been used for investigating how different levels of nutrients and drugs influence the development and the metabolic phenotypes of emerging Drosophila embryos [45], our data are model behavior of oviparous animals, in which development occurs outside the mother’s body. In mammals, maternal HDACi cross the placenta and affect embryogenesis; hence any future intervention should be envisaged after birth when the mammalian central nervous system is still developing.

Lack of an anoxic compartment in the *Drosophila* gut shapes a microaerophilic microbiota constituted, in laboratory strains, by few genera [46]. The most abundant taxa are *Firmicutes*, mainly *Lactobacillus* spp. and *alpha-Proteobacteria*, mainly *Acetobacteraceae*. Despite evolutionary divergence with the human microbiota, recent studies showed that an altered relative abundance of these genera can result in gut homeostasis disturbance [47], growth delay [48], and behavioral changes [49]. Considering the results obtained from microbiota analysis of RSTS patients, we analyzed the microbial community of heterozygous *nej* insects compared to control animals. Results showed that, accounting for the species-specificity of the microbiota, the differences observed in RSTS patients compared to healthy siblings are recapitulated in RSTS flies compared to control animals, suggesting that patients–microbiota interactions could be modeled in *Drosophila*.

Overall, our results are in line with other studies [12,20,50], making a strong case for HDACi drug repurposing for future RSTS therapy. We envisage that the use of *Drosophila melanogaster* as an information rich in vivo RSTS model will speed up the transition from preclinical studies to clinical practice.

## 4. Materials and Methods

### 4.1. Cell Cultures

Lymphoblastoid cell lines (LCLs) from eight different RSTS patients (four carrying *CREBBP* mutations and four carrying *EP300* mutations, listed in Appendix A) [20,51,52,53] and seven healthy donors were obtained in collaboration with the Gaslini Genetic Bank service (Telethon Network of Genetic Biobanks); their use was approved by Ethics Committee of Università degli Studi di Milano (Comitato Etico number 99/20, 17 November 2020). Cells were maintained in RPMI 1640 culture medium supplemented with L-glutamine (Euroclone, Pero, Italy), 20% fetal bovine serum (Euroclone, Pero, Italy), and penicillin/streptomycin (Euroclone, Pero, Italy), and cultured in an incubator with 5% CO_2_ at 37 °C.

LCLs were exposed to four different HDAC inhibitors: Trichostatin A (TSA) (sc-3511, Santa Cruz Biotechnology, Dallas, TX, USA), Suberoylanilide hydroxamic acid (SAHA) (MK0683, Selleckchem, Houston, TX, USA), Valproic acid (VPA) (P4543, Sigma Aldrich, St. Louis, MO, USA), and Sodium Butyrate (NaB) (B5887, Sigma-Aldrich, St. Louis, MO, USA). We tested three different concentrations for each HDACi (Appendix A) [54,55,56,57,58,59] and selected the maximum dose and timing of exposure, ensuring acceptable LCLs survival (data not shown). Cells were incubated with vehicles (H_2_O or DMSO) at the maximum time (24 h), TSA 2 µM for 2 h, SAHA 2 µM for 24 h, VPA 2 mM for 24 h, or NaB 5 mM for 24 h as suggested from the literature (Appendix A). Data were normalized on untreated cells and in vehicles for accounting for proliferation rate differences in basal condition between HD and RSTS lines.

### 4.2. AlphaLISA^®^ Assay

After treatments, lymphoblastoid cellular pellets were obtained by centrifugation and frozen at −80 °C. An amount of 10,000 cells/well resuspended in 60 µL of culture media was used in order to perform AlphaLISA^®^ assay (PerkinElmer, Waltham, MA, USA) according to the manufacturer’s protocol. Briefly, cells were incubated 15 min with Cell-Histone Lysis buffer and 10 min with Cell-Histone Extraction buffer; 30 µL of lysates were incubated with 10 µL of Acceptor mix 1h at room temperature (RT) and then 10 µL of Donor mix was added overnight at RT. Replicates were tested with both AlphaLISA Acetylated-Histone H3 Lysine 27 (H3K27ac) Cellular Detection Kit (AL720, PerkinElmer, Waltham, MA, USA) and AlphaLISA unmodified Histone H3 Lysine 4 (H3K4) Cellular Detection Kit (AL719, PerkinElmer, Waltham, MA, USA) for normalization. PerkinElmer EnSight™ plate reader was used for the detection of the chemiluminescent signal.

### 4.3. Ki67 and TUNEL Assay

After treatments, at least 1.5 × 10^4^ LCLs were seeded in duplicate on SuperFrost Plus slides (Thermofisher Scientific, Waltham, MA, USA) through 5 min of cytospin at 500 rpm, followed by 10 min of incubation with PFA 4% and washed. Slides were stored at 4 °C until Ki67 or TUNEL assays were performed.

Briefly, for Ki67 assay slides, samples were put in a wet chamber and cells permeabilized with PBT buffer (Phosphate-Buffered Saline (PBS) with 0.2% Triton) for 10 min at room temperature (RT); blocking of non-specific sites was obtained by slide incubation with PBT supplemented with 10% FBS for 30 min at RT. Slides were first incubated overnight at 4 °C with the anti-Ki67 antibody (#9129 Cell Signaling, Danvers, MA, USA, 1:400), washed with PBT, and then incubated with Alexa-488 anti-Rabbit secondary antibody (#6441-30 SouthernBiotech, Birmingham, AL, USA, 1:250) for 2 h. Slides were washed with PBT and water, mounted with EverBrite Mounting Medium with DAPI (23002, Biotium, Landing Parkway Fremont, CA, USA), and fluorescent microscopic images of proliferative cells (Ki67+) were acquired and analyzed with ImageJ software (National Institute of Health, Bethesda, MD, USA).

Terminal deoxynucleotidyl transferase (TdT) dUTP Nick-End Labeling (TUNEL) assay was performed using In Situ Cell Death Detection kit, AP (Roche Diagnostics, Basilea, Switzerland), in order to detect apoptotic cells, according to manufacturer’s protocol. Cells, previously seeded on slides were incubated with a permeabilization solution (0.1% Triton 100X and 0.1% sodium citrate) for 2 min at 4 °C, then washed with PBS and incubated with TUNEL mixture (composed by Enzyme Solution added to Label Solution) in a wet chamber for 1h at 37 °C. After 3 PBS washes, slides were incubated with Converter AP for 30 min at 37 °C and then with Substrate Solution (2% NBT/BCIP stock solution in NBT/BCIP Buffer) for 10 min at RT and dark. Finally, following PBS washes, mounted with DABCO mounting medium and brightfield microscopic images of apoptotic cells (TUNEL+) were acquired and analyzed with ImageJ software (National Institute of Health, Bethesda, MD, USA).

Both fluorescent and brightfield slide images were acquired by NanoZoomer S60 Digital Slide Scanner (Hamamatsu Photonics, Hamamatsu City, Japan) at 20× and 80× magnification, and two randomly selected fields for each experimental group at 20× were selected for blinded cells counts by three different operators. Panel images of Ki67+ cells were instead acquired by confocal microscopy A1/A1R (Nikon Corporation, Tokyo, Japan) at 60× and 100× magnification. The number of Ki67+ and TUNEL+ cells was normalized on the total cell number per image.

### 4.4. Subject Recruitment and Sampling for Gut Microbiota Profiling

For this study, 23 RSTS subjects and 16 healthy siblings were enrolled. All subjects were recruited in collaboration with the Italian family RSTS association “Associazione RTS Una Vita Speciale ONLUS”.

For both patients and controls, exclusion criteria were treatments with antibiotic and/or probiotic/prebiotic assumption during the previous 3 months. For RSTS patients, inclusion criteria were confirmed clinical diagnosis with (20/23) or without (3/23) demonstrated *CREBBP*/*EP300* mutation. RSTS diagnosis of all patients was confirmed by an expert geneticist (DM) and genetic tests were performed in our laboratory (CG).

In conjunction with the stool sample collection, a 3 day dietary survey (preceding the sample collection) was filled by caregivers. Dietary food records were processed using commercially available software (ePhood V2, Openzone, Bresso, Italy).

The study was approved by the Ethics Committee of San Paolo Hospital in Milan (Comitato Etico Milano Area 1, Protocol number 2019/EM/076, 2 May 2019); written informed consent was obtained from enrolled subjects or caregivers.

### 4.5. Bacterial DNA Extraction and 16S rRNA Gene Sequencing of Human Gut Microbiota

Bacterial genomic DNA in stool samples was extracted as previously described [60] by using the Spin stool DNA kit (Stratec Molecular, Berlin, Germany), according to the manufacturer’s instructions. Briefly, after homogenizing fecal samples in the lysis buffer for inactivating DNases, Zirconia Beads II were added for a complete lysis of bacterial cells by using TissueLyser LT. Bacterial lysates were then mixed with InviAdsorb reagent, a step designed to remove PCR inhibitors. Bacterial DNA was eventually eluted in 100 μL of buffer. Then, 25 ng of extracted DNA was used to construct the sequencing library. The V3–V4 hypervariable regions of the bacterial 16S rRNA were amplified with a two-step barcoding approach according to the Illumina 16S Metagenomic Sequencing Library Preparation (Illumina, San Diego, CA, USA). Library quantification was determined using the DNA High Sensitivity Qubit kit (Thermofisher Scientific, Waltham, MA, USA) and Agilent 2100 Bioanalyzer System (Agilent, Santa Clara, CA, USA); libraries were pooled and sequenced on a MiSeq platform (Illumina, San Diego, CA, USA) in a 2 × 250 bp paired-end run. Obtained 16S rRNA gene sequences were analyzed using PANDAseq [61], and low-quality reads were filtered and discarded. Reads were then processed using the Quantitative Insights Into Microbial Ecology (QIIME) pipeline (release 1.8.0) [62] and clustered into Operational Taxonomic Unit (OTUs) at 97% identity level and discarding singletons (i.e., OTUs supported by only 1 read across all samples) as likely chimeras. Taxonomic assignment was performed via the Ribosomal Database Project (RDP) classifier [63] against the Greengenes database (version 13_8; ftp://greengenes.microbio.me/greengenes_release/gg_13_8_otus, accessed on 22 February 2021), with a 0.5 identity threshold. Alpha-diversity was computed using the Chao1, the number of OTUs, Shannon diversity, and Faith’s Phylogenetic Diversity whole tree (PD whole tree) metrics throughout the QIIME pipeline. Beta-diversity was assessed by weighted and unweighted UniFrac distances [64] and principal coordinates analysis (PCoA).

### 4.6. Fecal Short-Chain Fatty Acid Quantification

Concentrations of acetate, propionate, iso-butyrate, butyrate, and iso-valerate were assessed according to Bassanini et al. [65]. The measurement of SCFAs was performed by gas chromatography, using a Varian model 3400 CX Gas chromatograph fitted with FID detector, split/splitless injector, and a SPB-1 capillary column (30 m × 0.32 mm ID, 0.25 μm film thickness; Supelco, Bellefonte, PA, USA). Calibration curves of SCFAs in concentration between 0.25 and 10 mM were constructed to obtain SCFAs quantification, and 10 mM 2-ethylbutyric acid was used as an internal standard. Results are expressed as mg/g of dry weight of feces.

### 4.7. Drosophila Melanogaster Stocks and HDACi Feeding

The following fly strains were used in this study: *Drosophila yw* strain used as control and *w[*] P{w[+mC]=lacW}nej[P]/FM7c* known as *nejire* (*nejP/+*) mutant strain (#3728; Bloomington Drosophila Stock Center, Bloomington, IN, USA). *nejP* contains a P-element 347bp upstream of the second exon of *nej* gene and behaves as a loss of function mutant [27]. Flies were maintained and raised into vials containing a standard food medium composed of yeast, cornmeal, molasses, agar, propionic acid, tegosept, and water. All the strains were kept at 25 °C. To prepare food with HDACi, stock solutions of VPA (1 and 2.5 mM) and NaB (10 and 20 mM) solutions were diluted 1:10 in the food before solidification but under 65 °C to prevent heat damage of the compounds.

### 4.8. Fly Treatment and Embryo Immunostaining

To identify homozygous animals in sibling crosses, *nej* mutants were balanced over *FM7, kr-GAL4 UAS-GFP* chromosome. Adult *nej* mutant flies were life cycle-synchronized and treated with mock, VPA (2.5 mM) or NaB (20 mM) for four days. The drugs were mixed with standard food as described above at room temperature. For egg collection, adult flies were placed in cages for 4 h and then removed. Fertilized eggs were collected after 8h and stained as previously described [66]. Briefly, embryos were collected, dechorionated and fixed with a mixture of 4% paraformaldehyde and heptane. Following washes, embryos were permeabilized and blocked with PBT (PBS containing 0.1% Triton X-100 and 1% BSA) for 3 h. Staining was performed overnight with primary mouse anti-wg 1:50 (4D4, Developmental Studies Hybridoma Bank, Iowa City, IA, USA). Secondary goat anti-mouse-Cy3 (1:500) was used for 2 h. Images were taken with a Nikon AR1 confocal microscope using a 10X objective (Nikon Corporation, Tokyo, Japan).

### 4.9. Bacterial DNA Extraction and 16S rRNA Gene Sequencing of Drosophila Melanogaster Gut Microbiota

For microbiota sequencing, adult fly guts were dissected to avoid environmental contamination. Briefly, flies were anesthetized in ice for 5 min in a Petri dish and transferred, one by one, to the dissection dish and immersed in 50 μL-drop of cold PBS. First, wings and legs were removed. Surgical forceps were used to gently separate the insect head from the body and exposing the foregut. The abdominal cuticle was then cut and dissected out, and the hindgut pulled outside the abdominal cavity. To completely free the entire gastrointestinal tract, the insect head and Malpighian tubes were removed. Only undamaged organs were further processed. Dissected guts, three per experimental condition, were immediately transferred into vials containing 100 µL of cold PBS and kept at −80 °C until use. A total of 15 *nej* and 15 *yw* flies were processed.

Bacterial DNA was extracted by means of QIAamp DNA Microbiome Kit (Qiagen, Hilden, Germany), designed to achieve enrichment of bacterial DNA from low biomass samples. Briefly, the depletion of host cells was performed by adding lysis buffer and benzonase to samples. Bacterial cell lysis was carried out by bead beating in the TissueLyser LT instrument (Qiagen, Hilden, Germany). Lysates were transferred to QIAamp UCP Mini Columns and processed according to the manufacturer’s instructions. DNA was eluted in 30 µL of the provided buffer. Library preparation and sequencing were performed as described above for human samples on the Illumina platform.

### 4.10. Statistical Analysis

Biological cell data were analyzed using Prism software (GraphPad Software, Sand Diego, CA, USA) and expressed as mean ± Standard Deviation (SD). Student’s t-tests were used to compare means between groups in AlphaLISA, Ki67, and TUNEL assays (LCLs acetylation, proliferation, and death rate), and in phenotypic evaluation (*Drosophila* embryo survival), with *p* < 0.05 considered significant (* *p* < 0.05; ** *p* < 0.01; *** *p* < 0.001 for graphics relative to in vitro model); the correlation between HDACi-induced acetylation and proliferative or apoptotic cells was calculated using Pearson correlation coefficient (−1 < r < 1) and Pearson correlation *p*-value, significant for *p* < 0.05.

For microbiota analysis, statistical evaluation among alpha-diversity indices was performed by a non-parametric Monte Carlo-based test, using 9999 random permutations. The PERMANOVA test (adonis function) in the R package vegan (version 2.0-10) was used to compare the microbial community structure of RSTS and HD subjects within the beta-diversity analysis. For evaluating differences in taxonomic relative abundances, the pairwise t-test from the package “rstatix” (version 0.6.0) in the RStudio software (version 1.2.1335; R version 3.6.3) was used. *p*-values < 0.05 were considered significant for each analysis.

## Figures and Tables

**Figure 1 ijms-22-03621-f001:**
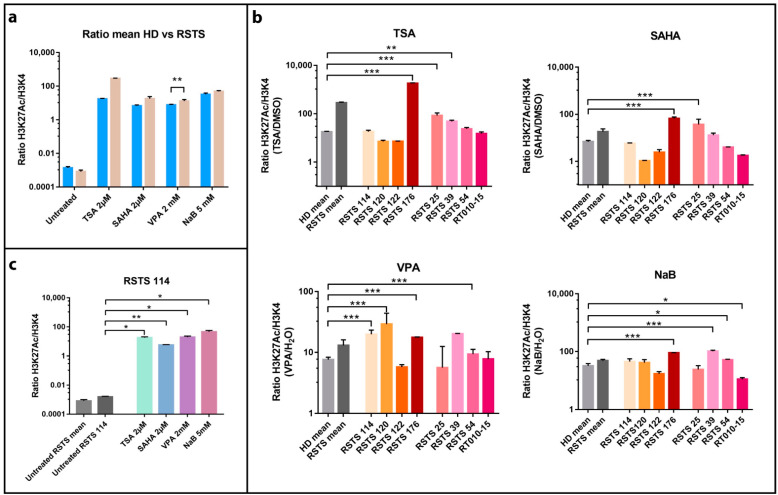
Histone acetylation on Rubinstein–Taybi syndrome (RSTS) lymphoblastoid cell lines (LCLs) upon acetyltransferases (HAT) and deacetylases (HDAC) inhibitors exposure. H3K27 acetylation levels normalized on H3K4 unmodified, assessed by AlphaLISA^®^; levels of acetylation upon HDAC inhibitors (HDACi) are expressed as a ratio between the treatment and respective vehicle (HDACi/vehicle); on the Log scale, *Y*-axis H3K27 acetylation levels normalized, on *X*-axis lists of epigenetic treatments or untreated/treated single LCL or LCLs means. (**a**) Means of values of H3K27 acetylation in healthy donors (HD, in blue) and patients LCLs (RSTS, in pale brown) untreated and exposed to four different HDACi (trichostatin A (TSA) 2 µM, suberoylanilide hydroxamic acid (SAHA) 2 µM, valproic acid (VPA) 2 mM, and sodium butyrate (NaB) 5 mM). (**b**) H3K27 acetylation in eight RSTS LCLs (*CREBBP* LCLs in shades of red, *EP300* LCLs in shades of pink) after exposure with the four different HDACi, compared to treated HD and RSTS means. (**c**) Insight on the single-patient response (RSTS 114) to the four compounds compared to untreated RSTS means and RSTS 114. Groups were compared using Student’s *t*-test as statistical method (* *p* < 0.05; ** *p* < 0.01; *** *p* < 0.001).

**Figure 2 ijms-22-03621-f002:**
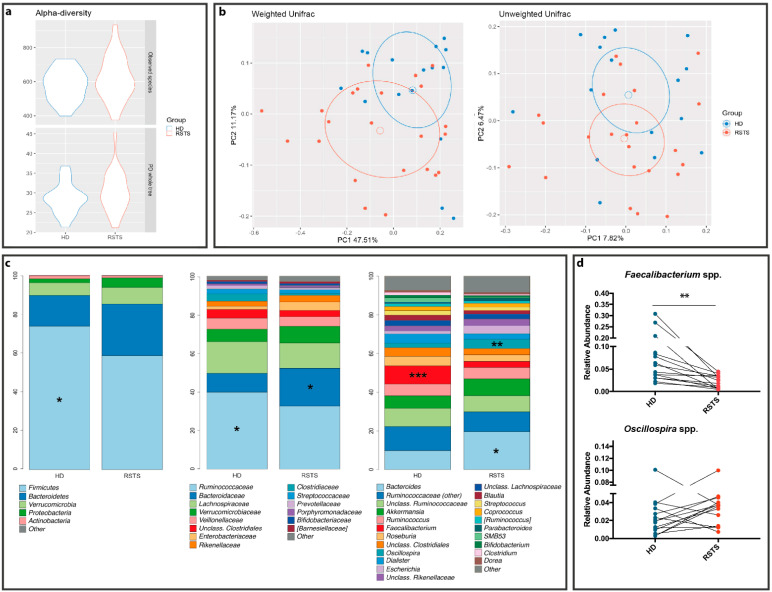
RSTS gut microbiota analysis. (**a**) alpha diversity. The violin plot shows biodiversity values for observed species and Faith’s phylogenetic metrics. No statistically relevant differences were seen. (**b**) Principal coordinate analysis (PCoA) according to weighted and unweighted Unifrac distances. Microbial communities are statistically different (*Adonis* test: unweighted *p* = 0.019; weighted *p* = 0.023). The first and second principal coordinates are shown in the plot for both distances. (**c**) Bacterial composition of HD and RSTS groups. Relative taxonomic abundances are shown at phylum, family, and genus phylogenetic levels. All bacterial taxa present at <1% relative abundance were grouped into the “Other” classification. ***: *p* < 0.005; **: *p* < 0.01; *: *p* < 0.05. (**d**) *Faecalibacterium* spp. and *Oscillospira* spp. relative abundances (both significantly different between RSTS and HD) were compared within matched family members (patient/sibling, *n* = 16). For *Oscillospira* we did not observe a common pattern; *Faecalibacterium* spp. was significantly reduced in RSTS (*p* = 0.0021, Wilcoxon signed-rank test).

**Figure 3 ijms-22-03621-f003:**
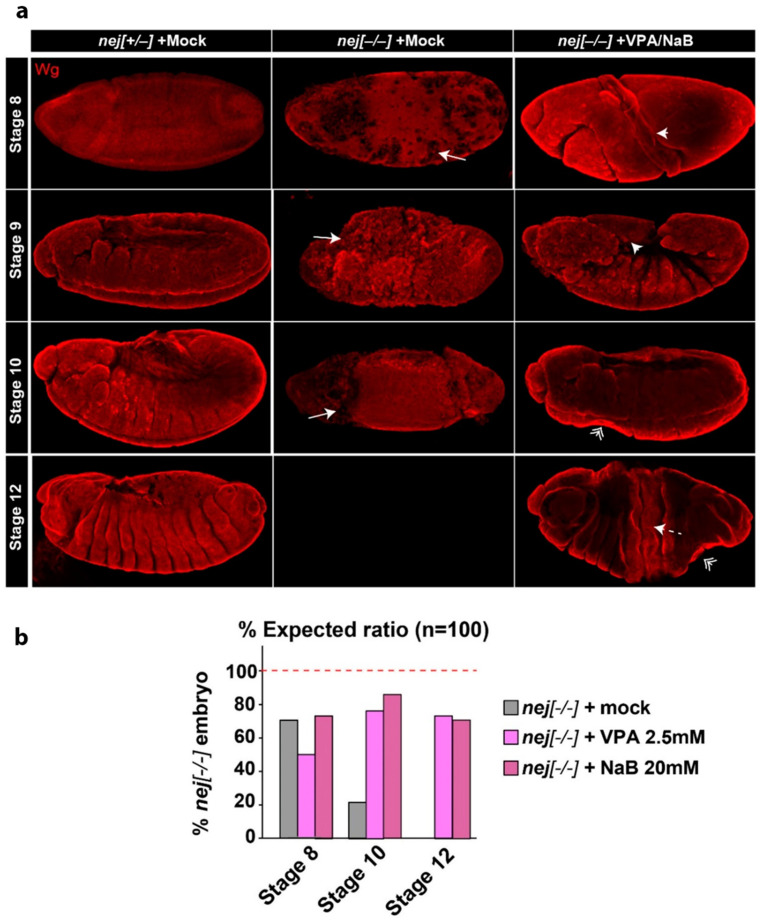
Developmental and morphological defects in *nej* mutant embryos are partially ameliorated by VPA or NaB treatment. (**a**) wingless (wg) staining of *nej* mutant embryos treated as indicated. Representative projections of confocal imaging from stages 8 to 12 are shown. The defects detected in *nej* mutant embryos with or without treatment include loss or uneven wg staining (Drilled, arrow), twisting (arrowhead), presence of bottlenecks (double-arrowheads), or cracks (dashed-arrow). (**b**) Quantification of embryo survival and of the above phenotypes in *nej* mutant embryos at stages 8 to 12. VPA or NaB treatments partially rescue embryo development, allowing *nej* mutant embryos to survive beyond stage 8 although with aberrant phenotypes, with significant embryo survival at stage 10 compared to untreated embryos (*p* > 0.01). Student’s t-test was used as statistical method for comparing *nej* groups survival, with *p* < 0.05 considered significant.

**Figure 4 ijms-22-03621-f004:**
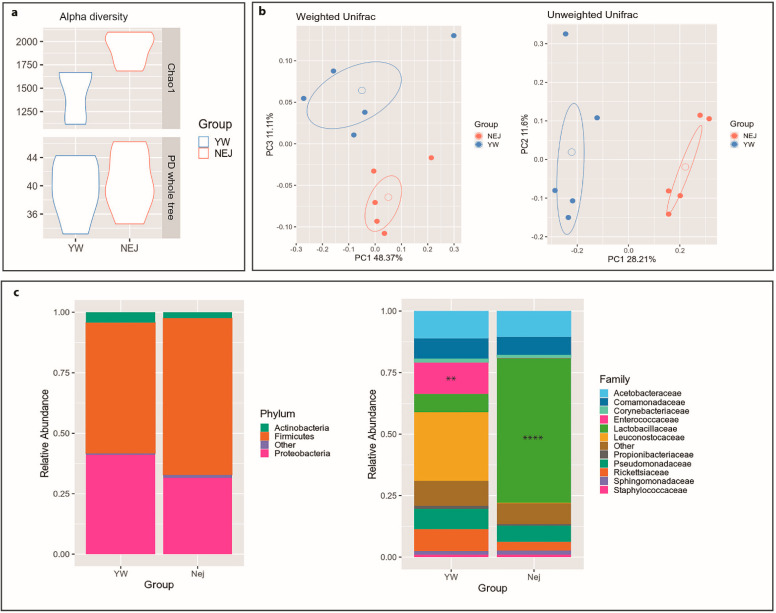
*Drosophila* gut microbiota analysis. Gut microbial communities were characterized by 16S rRNA gene sequencing. (**a**) Alpha diversity. Violin plots show biodiversity values for Chao1 (*p* = 0.005) and Faith’s phylogenetic metrics (PD whole tree; *p* = 0.668). (**b**) Principal coordinate analysis (PCoA) for both Unifrac distances. (*Adonis* test: weighted *p* = 0.216; unweighted *p* = 0.007). (**c**) Relative taxonomic abundances of the gut bacterial composition in *yw* and *nej* flies at phylum and family levels. All bacterial taxa present at <1% relative abundance were grouped into the “Other” classification. ****: *p* < 0.001; **: *p* < 0.01.

## Data Availability

Sequencing data of 16S rRNA amplicons have been deposited in NCBI Short-Read Archive (SRA) under accession number PRJNA616211 (http://www.ncbi.nlm.nih.gov/bioproject/PRJNA616211, accessed on 22 February 2021).

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
