# Peer review of "Insights into the Role of the Microbiota and of Short-Chain Fatty Acids in Rubinstein–Taybi Syndrome"

_ijms, 2021, doi:10.3390/ijms22073621_

Round 1

Reviewer 1 Report

Various benign and malignant tumors have been described in RSTS individuals.

1) Can you comment on this point?

2) Have you observed whether HDACi therapy also allows regression of these tumors?

Author Response

Questions

Various benign and malignant tumors have been described in RSTS individuals.

1) Can you comment on this point?

2) Have you observed whether HDACi therapy also allows regression of these tumors?

Answers

We thank the reviewer for pointing out this issue, and we report below our amendments:

1) we added a brief summary of the current state of the art about tumors in RSTS patients (line 64);

2) we found some interesting results on HDACi use on tumors that we added in Discussion (line 270)

Reviewer 2 Report

Manuscript ID: ijms-1138293
Type of manuscript: Article
Title: Insights into the role of the microbiota and of short chain fatty acids in Rubinstein-Taybi syndrome
Authors: Elisabetta Di Fede, et al.

In this report, Fede et al. present evidence that butylate which is produced by microbiota and acts as a potent histone deacetylase inhibitor (HDACi), is potentially lower in fecal samples of Rubinstein-Taybi syndrome (RSTS) patients.   RSTS is caused by a gene mutation in CBP and p300, lysine acetyltransferases.    The authors used an in vivo model of the “nejire (nej)” drosophila mutant, which is a CBP homolog in Drosophila, to study the effect of HDACi on the alleviate of RSTS phenotype.  When nejire mothers were fed with HDACi, abnormalities were rescued in the embryos .   These findings are interesting, however, some additional data are required, and the discussion part needs to be improved.

  1. In the title, the authors pointed out that the content of short chain fatty acids (SCFAs) produced by commensal microbiota is lower in RSTS patients.However, RSTS lymphoblastoid cells were treated with synthetic HDACi.  Why were SCFAs never used? This would be a further direct evidence that SCFAs change the acetylation level of histone.
  2. The maternal fly model of nej-/- was interesting.Does it mean that HDACi taken from the mother's mouth reaches to the egg? Transfering this model to mammals, does the maternal HDACi cross the placenta and affect the embryogenesis? This needs to be discussed in the Discussion part.
  3. Why do RSTS patients have less Faecalibacteria in their microbiota?Since HDACi has been already used for therapeutic treatment of several cancers, does HDACi treatment in RSTS patients cause enhanced Faecalibacterium content? 

Author Response

Q1

In this report, Fede et al. present evidence that butylate which is produced by microbiota and acts as a potent histone deacetylase inhibitor (HDACi), is potentially lower in fecal samples of Rubinstein-Taybi syndrome (RSTS) patients.   RSTS is caused by a gene mutation in CBP and p300, lysine acetyltransferases.    The authors used an in vivo model of the “nejire (nej)” drosophila mutant, which is a CBP homolog in Drosophila, to study the effect of HDACi on the alleviate of RSTS phenotype.  When nejire mothers were fed with HDACi, abnormalities were rescued in the embryos .   These findings are interesting, however, some additional data are required, and the discussion part needs to be improved.

In the title, the authors pointed out that the content of short chain fatty acids (SCFAs) produced by commensal microbiota is lower in RSTS patients.However, RSTS lymphoblastoid cells were treated with synthetic HDACi.  Why were SCFAs never used? This would be a further direct evidence that SCFAs change the acetylation level of histone.

A1

We thank the reviewer for these recommendations. In this work we tested different HDACi and among them, we used two short chain fatty acids, valproic acid and butyrate. Interestingly, the latter is a strong HDACi which is also produced by commensal microorganisms. To better delineate this point we clarified it (line 84).

Q2

The maternal fly model of nej-/- was interesting.Does it mean that HDACi taken from the mother's mouth reaches to the egg? Transfering this model to mammals, does the maternal HDACi cross the placenta and affect the embryogenesis? This needs to be discussed in the Discussion part.

A2

We thank the reviewer for raising the interesting point. Drosophila produces very large eggs and it is possible that changes in the nutrients available to the mothers affect egg formation. Considering that Drosophila is oviparous (development occurs outside the mother’s body), oogenesis is the unique step at which nutrients from the mother can influence embryogenesis. Thus, while the suggestion of the reviewer is fascinating the data from the Drosophila model are unlikely to shed light on possible transplacental transport. To clarify, we have added to Discussion a sentence on this matter (line 311).

Q3

Why do RSTS patients have less Faecalibacteria in their microbiota?

A3

We thank the reviewer for this question. Although there are no other studies on microbiota of RSTS patients to the best of our knowledge, we discuss other studies on genetic disorders that highlight a possible shared microbiota signature (line 287).

Q4

Since HDACi has been already used for therapeutic treatment of several cancers, does HDACi treatment in RSTS patients cause enhanced Faecalibacterium content?

A4

We are grateful to the reviewer for this comment. To our knowledge, studies on microbiota changes in tumor models upon HDACi treatment are scarce, thus it would be interesting to further investigate this relationship. To date, dysbiosis and mainly loss of Faecalibacterium genus characterizes different pathological conditions, representing an indicator for gut and human health (Ferreira-Halder et al 2017, DOI: 10.1016/j.bpg.2017.09.011). In addition, butyrate was shown to be the major end product of Faecalibacterium metabolism, specifically able to exert anti-inflammatory effects through its HDACi activity (Zhou et al 2018, DOI: 10.1093/ibd/izy182).

Round 2

Reviewer 2 Report

The authors answered all of the questions by the reviewer.